# Meta-Learning under Task Shift

**Lei Sun**                                                                *sun.lei.sn5@is.naist.jp*
*Nara Institute of Science and Technology*

**Yusuke Tanaka**                                                           *ysk.tanaka@ntt.com*
*NTT Communication Science Laboratories*

**Tomoharu Iwata**                                                          *tomoharu.iwata@ntt.com*
*NTT Communication Science Laboratories*

**Reviewed on OpenReview:** *https://openreview.net/forum?id=0xV75W9OFN*

## Abstract

A common assumption in meta-learning is that meta-training and meta-test tasks are drawn from the same distribution. However, this assumption is often not fulfilled. Under such task shift, standard meta-learning algorithms do not work as desired since their unbiasedness is no longer maintained. In this paper, we propose a new meta-learning method called Importance Weighted Meta-Learning (IWML), which preserves unbiasedness even under task shift. Our approach uses both labeled meta-training datasets and unlabeled datasets in tasks obtained from the meta-test task distribution to assign weights to each meta-training task. These weights are determined by the ratio of meta-test and meta-training task densities. Our method enables the model to focus more on the meta-training tasks that closely align with meta-test tasks during the meta-training process. We meta-learn neural network-based models by minimizing the expected weighted meta-training error, which is an unbiased estimator of the expected error over meta-test tasks. The task density ratio is estimated using kernel density estimation, where the distance between tasks is measured by the maximum mean discrepancy. Our empirical evaluation of few-shot classification datasets demonstrates a significant improvement of IWML over existing approaches.

## 1 Introduction

The ability to learn quickly from a limited number of examples is a major distinction between humans and artificial intelligence (AI). Humans can acquire new tasks rapidly, leveraging their past learning experiences. On the other hand, most modern AI methodologies are trained specifically for a single task, which restricts their utility to that particular task alone. Therefore, when faced with new tasks, it is required to newly gather a huge amount of data for training. However, collecting sufficient training data is often expensive and time-consuming, presenting a considerable challenge in real-world applications. Meta-learning, a widely used approach adopted to address this challenge, employs the transferable knowledge gained from meta-training tasks to boost the learning efficacy for meta-test tasks (Vilalta & Drissi, 2002; Nichol et al., 2018; Vanschoren, 2018; Santoro et al., 2016; Khodak et al., 2019). When formulating a meta-learning approach, it is generally assumed that the meta-training and meta-test tasks are drawn from the same distribution (Yoon et al., 2018; Li et al., 2017; Rajeswaran et al., 2019; Yin et al., 2019; Zintgraf et al., 2019). This assumption, however, is frequently unfulfilled. This mismatch hampers our ability to learn transferable knowledge that could be beneficial for meta-test tasks, thereby leading to erroneous predictions. We refer to this phenomenon, where meta-training and meta-test tasks follow different distributions, as 'task shift'.

Standard meta-learning algorithms (Finn et al., 2017; Snell et al., 2017; Garnelo et al., 2018) optimize model parameters by minimizing the expected meta-training error. However, these methods fall short under task shifts, as the expected meta-training error does not provide an unbiased estimate of the expected meta-test

error. To address task shifts, a method called Weighted Meta-Learning was proposed by Cai et al. (2020). This approach aims to find weights for each meta-training task to minimize the distance between the meta-training tasks and a single meta-test task. However, it assumes that there is only one meta-test task, requires labeled data from tasks that appear in the meta-test phase for weight calculation, and is limited to regression scenarios. These limitations hinder its application in real-world settings. Additionally, task augmentation strategies (Rajendran et al., 2020; Yao et al., 2021a; Wu et al., 2022) aim to generate tasks that are not provided during the meta-training phase to improve performance on meta-test tasks. Yet, in the case of task shifts, the newly generated tasks follow the meta-training task distribution and fail to enhance performance on tasks from the meta-test task distribution.

In this paper, we introduce a new meta-learning approach called Importance Weighted Meta-Learning (IWML), designed to reduce misestimation caused by task shift. Our method assigns weights to each meta-training task, reflecting the *importance* of a meta-training task for learning meta-test tasks. This weight is determined by the ratio of meta-test and meta-training task densities. Our method enables the model to focus more on the meta-training tasks that closely align with meta-test tasks during the meta-training process. In the proposed method, we meta-learn neural network-based models by minimizing the expected weighted meta-training error, which is an unbiased estimator of the expected error over meta-test tasks. We estimate these densities using labeled meta-training datasets and unlabeled datasets in tasks obtained from the meta-test task distribution. The task density ratio is estimated using Kernel Density Estimation (KDE), where the distance between tasks is measured by the Maximum Mean Discrepancy (MMD). Figure 1 illustrates the meta-training procedure of IWML.

The proposed method is applicable in situations where we have datasets from multiple tasks that are related to the meta-training tasks, but they follow different distributions. For example, consider an image classification problem in different scenarios where the meta-training datasets consist of many tasks composed of images taken by ordinary cameras, and the meta-test datasets consist of many tasks composed of images taken under a microscope. Even though the images come from different distributions, they are related.

The following are the main contributions of this paper:

- We propose a new meta-learning method that provides an unbiased estimator of the expected error over meta-test tasks, even under a task shift.

- We propose a task density estimation method by leveraging KDE in conjunction with the MMD distance.

- To the best of our knowledge, our work represents the first effort to tackle the issue of task shift in meta-learning by leveraging unlabeled datasets from tasks derived from the meta-test task distribution. Furthermore, our experimental results demonstrate that our method can accurately estimate reasonable weights, even when the tasks utilized for weight computation are different from those in the meta-test phase.

- We empirically demonstrate that our proposed method outperforms existing meta-learning algorithms, the weighted meta-learning method, and state-of-the-art task augmentation strategies under a task shift.

## 2 Related works

### 2.1 Covariate shift

The covariate shift problem is a common issue in conventional supervised learning scenarios (Shimodaira, 2000). This problem arises when supervised learning algorithms fail to provide an unbiased estimator because the feature vectors used for training and test come from different distributions, while the conditional distributions of the output given the feature vectors remain unchanged.

The covariate shift problem is related to the task shift problem in that both involve differences in distributions between the training and test phases. However, the covariate shift problem occurs within a single task, where

the distribution of feature vectors shifts between the training and test phases. In contrast, the task shift problem in meta-learning refers to a shift in the distribution of tasks themselves between the meta-training and meta-test phases. Various methods have been developed to address the covariate shift problem by estimating weights and applying them to the training feature vectors (Sugiyama et al., 2007; Bickel et al., 2009; Sugiyama et al., 2008). However, these methods are inapplicable to task shift. Although these methods and the proposed method both use weights to calculate an unbiased error, they are only capable of computing weights for each sample within a single task. In cases where it is necessary to compute weights for each task, such as in task shift situations, these methods are inapplicable.

## 2.2 Unsupervised domain adaptation

Although the methods developed to address Unsupervised Domain Adaptation (UDA) and the proposed method both make use of unlabeled test datasets to improve performance, these UDA techniques are typically designed for scenarios involving only two tasks: training and test. As such, they are not adaptable to meta-learning frameworks, which involve multiple meta-training and meta-test tasks (Finn et al., 2017; Yao et al., 2019; Snell et al., 2017).

UDA aims to transfer knowledge from a labeled source domain to an unlabeled target domain. The dominant approaches for UDA focus on acquiring domain-invariant features and can be categorized into two principal strategies. The first strategy aims to explicitly minimize domain discrepancies by utilizing measures such as Maximum Mean Discrepancy (MMD) (Long et al., 2015; 2017; Borgwardt et al., 2006) to evaluate domain similarities. Meanwhile, other studies introduce metrics based on second-order or higher-order statistical differences (Sun et al., 2016; Sun & Saenko, 2016). An alternative approach employs adversarial training to develop domain-invariant features (Saito et al., 2018; Sankaranarayanan et al., 2018), a method pioneered by the work on Domain-Adversarial Neural Networks (DANN) (Ganin & Lempitsky, 2015; Ganin et al., 2016). This technique involves training a domain discriminator to distinguish between source and target domains while concurrently training a feature extractor to deceive the discriminator, leading to the alignment of features.

## 2.3 Meta-learning

A number of frameworks have been proposed for meta-learning (Hospedales et al., 2021), including gradient-based (Finn et al., 2017; Li et al., 2017; Yoon et al., 2018; Finn et al., 2018), metric-based (Snell et al., 2017; Koch et al., 2015; Vinyals et al., 2016), and black box-based (Garnelo et al., 2018; Mishra et al., 2017; Hochreiter et al., 2001) methods. Among them, Model-agnostic Meta-Learning (Finn et al., 2017), Prototypical Networks Snell et al. (2017), and Conditional Neural Processes (Garnelo et al., 2018) serve as representative methods for each respective framework. These existing meta-learning methods assume that meta-training and meta-test tasks adhere to the same distribution, and they optimize model parameters by minimizing the expected meta-training error. In Section 3.2, we explain that in cases of task shift, due to inconsistencies between expected meta-training and meta-test errors, optimizing model parameters by minimizing the expected meta-training error does not guarantee a reduction in meta-test error.

Yao et al. (2021b) have conducted a study on meta-learning and proposed a method called Adaptive Task Scheduler (ATS). Although both ATS and our method assign weights to each meta-training task, there are several differences between the two methods. The main difference between our proposed method and ATS lies in the problems they aim to solve. ATS introduces weights to optimize the generalization capacity of the model and to avoid meta-overfitting, whereas our proposed method addresses meta-learning under task shift. These are two distinct problems, and because of the differences in the problems they solve, the methods for calculating weights are also different. ATS calculates weights based on the loss on the query set and the gradient similarity between the support set and the query set. In contrast, our method calculates weights based on the ratio of meta-test and meta-training task densities. While ATS is effective in optimizing the generalization capacity of the model and avoiding overfitting, it is not designed to address task shift issues.

### 2.4 Weighted meta-learning

Cai et al. (2020) have conducted a study on meta-learning under task shift and proposed a method called Weighted Meta-Learning (WML). In WML, it is assumed that there is only one meta-test task, and weights are assigned to each meta-training task to minimize the distance between the single meta-test task and multiple meta-training tasks. While we also introduce a method that assigns weights to meta-training tasks, our approach differs from WML in several aspects. (1) WML assumes there is only one meta-test task, whereas our proposed method can consider multiple meta-test tasks by using the meta-test distribution. (2) WML requires recalculating the weights when applied to a new meta-test task, whereas our proposed method can be applied to unseen meta-test tasks that follow the meta-test task distribution without the need for weight recalculation. (3) WML necessitates labeled datasets from tasks involved in the meta-test phase for weight computation. In contrast, our proposed method allows for the use of unlabeled datasets from tasks that are drawn from the meta-test distribution but do not appear in the meta-test phase to compute weights. (4) WML is restricted to regression scenarios and constrained to the use of square and hinge loss as its loss functions. Conversely, our proposed method is not only applicable in regression scenarios but also extends to multi-classification scenarios, offering flexibility with no limitations on the choice of the loss function, parameter learning methods, and models.

### 2.5 Task augmentation strategies

The objective of task augmentation strategies in meta-learning is to generate tasks that are not provided during the meta-training phase, thereby enhancing the performance of unknown tasks that are presented during the meta-test phase. To accomplish this objective, several existing methods have been developed. For instance, Meta-Aug (Rajendran et al., 2020) enhances a task by adding random noise to the labels of both support and query sets, while Meta-Mix (Yao et al., 2021a) blends support and query examples within a task. Furthermore, Adversarial Task Up-sampling (ATU) (Wu et al., 2022) aims to train a network that up-samples tasks adversarially, generating tasks that align with the true task distribution.

These task augmentation methods can improve performance for unseen tasks by generating additional tasks in scenarios where meta-training and meta-test tasks share the same task distribution. However, when the meta-training and meta-test tasks follow different task distributions, the newly generated tasks, derived from the meta-training task distribution, fail to improve the performance on tasks from the meta-test task distribution.

## 3 Preliminary

### 3.1 Problem setting

In standard meta-learning, the model is trained and evaluated on episodes of few-shot learning tasks. During the meta-training phase, let us consider we have a collection of meta-training tasks $\{\mathcal{T}_t^{\mathsf{TR}}\}_{t=1}^{T^{\mathsf{TR}}}$ drawn from meta-training task distribution $p^{\mathsf{TR}}(\mathcal{T})$. Each task is associated with labeled meta-training dataset $\mathcal{D}_t^{\mathsf{TR}} = \{(\mathbf{x}_{tn}^{\mathsf{TR}}, y_{tn}^{\mathsf{TR}})\}_{n=1}^{N_t^{\mathsf{TR}}}$, where $\mathbf{x}_{tn}^{\mathsf{TR}} \in \mathbb{R}^D$ is the $D$-dimensional feature vector of the $n$-th instance in the $t$-th meta-training task, $y_{tn}^{\mathsf{TR}}$ is the corresponding label, $T^{\mathsf{TR}}$ is the number of meta-training tasks, and $N_t^{\mathsf{TR}}$ is the number of instances.

In the meta-test phase, we are given meta-test task $\mathcal{T}_t^{\mathsf{TE}}$ drawn from meta-test task distribution $p^{\mathsf{TE}}(\mathcal{T})$. associated with labeled dataset $\mathcal{S} = \{(\mathbf{x}_n, y_n)\}_{n=1}^{N_\mathcal{S}}$, which is called support set. Here, the number of instances $N_\mathcal{S}$ is small. The goal is to predict label $y$ for each feature vector $\mathbf{x}$, which is called a query, in the meta-test task.

### 3.2 Meta-learning

In standard meta-learning, the expected error over meta-training tasks $\{\mathcal{T}_t^{\mathsf{TR}}\}_{t=1}^{T^{\mathsf{TR}}}$ is calculated by:

$$R^{\mathsf{TR}} = \mathbb{E}_{\mathcal{T}_t^{\mathsf{TR}} \sim p^{\mathsf{TR}}(\mathcal{T})} \left[ \mathbb{E}_{(\mathcal{S},\mathcal{Q}) \sim \mathcal{D}_t^{\mathsf{TR}}}[L(\mathcal{Q}|\mathcal{S};\theta)] \right], \tag{1}$$

where $\mathbb{E}$ denotes the expectation, support set and query set are denoted as $\mathcal{S} = \{\mathbf{x}_n, y_n\}_{n=1}^{N_\mathcal{S}}$ and $\mathcal{Q} = \{\mathbf{x}_n, y_n\}_{n=1}^{N_\mathcal{Q}}$ which contains pair of feature vectors and labels randomly generated from associated datasets of each task $\{\mathcal{D}_t^{\mathsf{TR}}\}_{t=1}^{T^{\mathsf{TR}}}$,

$$L(\mathcal{Q}|\mathcal{S}; \theta) = \frac{1}{N_\mathcal{Q}} \sum_{\mathbf{x}, y \in \mathcal{Q}} \ell(y, f(\mathbf{x}, \mathcal{S}; \theta)), \tag{2}$$

is the loss on the query set given the support set, $N_\mathcal{Q}$ is the number of paired data in the query set, $\theta$ is parameters of model $f$, and $f(\mathbf{x}, \mathcal{S}; \theta)$ is the output of the model for input $\mathbf{x}$, adapted using support set $\mathcal{S}$. The loss function represented by $\ell(y, f(\mathbf{x}, \mathcal{S}; \theta))$ measures the discrepancy between true label $y$ at input feature vector $\mathbf{x}$ and its estimate $f(\mathbf{x}, \mathcal{S}; \theta)$. Similarly, the expected error over target tasks is calculated from associated support set $\mathcal{S}$ and query set $\mathcal{Q}$:

$$R^{\mathsf{TE}} = \mathbb{E}_{\mathcal{T}_t^{\mathsf{TE}} \sim p^{\mathsf{TE}}(\mathcal{T})} \left[ \mathbb{E}_{(\mathcal{S}, \mathcal{Q}) \sim \mathcal{D}_t^{\mathsf{TE}}} [L(\mathcal{Q}|\mathcal{S}; \theta)] \right]. \tag{3}$$

In the situation where $p^{\mathsf{TR}}(\mathcal{T}) = p^{\mathsf{TE}}(\mathcal{T})$, the expected meta-training error $R^{\mathsf{TR}}$ matches the expected error over meta-test tasks $R^{\mathsf{TE}}$. Therefore, optimizing the model parameters by minimizing $R^{\mathsf{TR}}$ ensures a corresponding reduction in $R^{\mathsf{TE}}$. However, when $p^{\mathsf{TR}}(\mathcal{T}) \neq p^{\mathsf{TE}}(\mathcal{T})$, referred to as task shift, expected meta-training error $R^{\mathsf{TR}}$ and expected error over meta-test tasks $R^{\mathsf{TE}}$ do not match: $R^{\mathsf{TR}} \neq R^{\mathsf{TE}}$. Consequently, general meta-learning methods are designed to minimize $R^{\mathsf{TR}}$ by optimizing model parameters $\theta$. However, in cases of task shift, this approach does not ensure that $R^{\mathsf{TE}}$ will be minimized as well.

## 4 Proposed method

### 4.1 Problem setting

In this paper, we consider a scenario where there is a task shift, meaning that the meta-training and meta-test tasks follow different probability distributions, but the supports of meta-training and meta-test task distributions are overlapped.

During the meta-training phase, consider we have a set of meta-training tasks $\{\mathcal{T}_t^{\mathsf{TR}}\}_{t=1}^{T^{\mathsf{TR}}}$, each drawn from $p^{\mathsf{TR}}(\mathcal{T})$. Additionally, we also have $T^{\mathsf{TE}}$ meta-test tasks obtained from $p^{\mathsf{TE}}(\mathcal{T})$, where each task is associated with unlabeled dataset $\mathcal{U}_t^{\mathsf{TE}} = \{(\mathbf{x}_{tn}^{\mathsf{TE}})\}_{n=1}^{N_t^{\mathsf{TE}}}$. Here, $\mathbf{x}_{tn}^{\mathsf{TE}}$ is the $D$-dimensional feature vector of the $n$-th instance in the $t$-th meta-test task, and $N_t^{\mathsf{TE}}$ is the number of instances. It is important to note that these meta-test tasks, associated with unlabeled data, may differ from those in the meta-test phase. The settings for the meta-test phase are consistent with standard meta-learning.

### 4.2 Importance Weighted Meta-Learning

To address task shift, we propose Importance Weighted Meta-Learning (IWML), a method that estimates model parameters $\theta$ by minimizing the following expected weighted meta-training error $R^{\mathsf{IWML}}$. This error is an unbiased estimator of the expected error over the meta-test task distribution:

$$R^{\mathsf{IWML}} = \mathbb{E}_{\mathcal{T}_t^{\mathsf{TR}} \sim p^{\mathsf{TR}}(\mathcal{T})} \left[ \mathbf{w}_t \mathbb{E}_{(\mathcal{S}, \mathcal{Q}) \sim \mathcal{D}_t^{\mathsf{TR}}} [L(\mathcal{Q}|\mathcal{S}; \theta)] \right], \tag{4}$$

where

$$\mathbf{w}_t = \frac{p^{\mathsf{TE}}(\mathcal{T}_t^{\mathsf{TR}})}{p^{\mathsf{TR}}(\mathcal{T}_t^{\mathsf{TR}})}, \tag{5}$$

indicates the importance of $\mathcal{T}_t^{\mathsf{TR}}$; it reflects how crucial this task is for solving meta-test tasks. $p^{\mathsf{TE}}(\mathcal{T}_t^{\mathsf{TR}})$ indicates the meta-test task density of meta-training task $\mathcal{T}_t^{\mathsf{TR}}$, while $p^{\mathsf{TR}}(\mathcal{T}_t^{\mathsf{TR}})$ represents its meta-training task density.

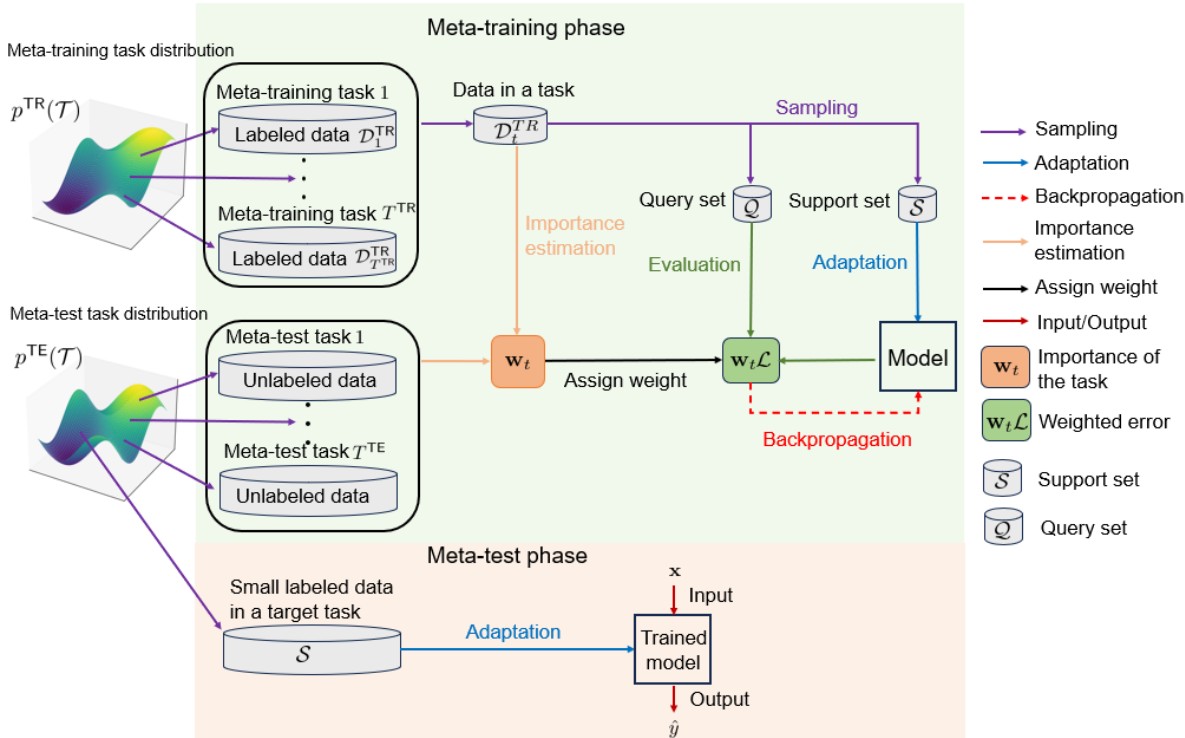

Figure 1: Illustration of the IWML algorithm. During the meta-training phase, our model is trained by minimizing the weighted meta-training error across various meta-training tasks. In the meta-test phase, the trained model predicts label $y$ for individual feature vector $\mathbf{x}$ using a support set.

Under the task shift, expected weighted meta-training error $R^{\mathsf{IWML}}$ and expected meta-test error $R^{\mathsf{TE}}$ match:

$$
\begin{aligned}
R^{\mathsf{IWML}} &= \int \frac{p^{\mathsf{TE}}(\mathcal{T}_t^{\mathsf{TR}})}{p^{\mathsf{TR}}(\mathcal{T}_t^{\mathsf{TR}})} \left[ \mathbb{E}_{(\mathcal{S},\mathcal{Q})\sim\mathcal{D}_t^{\mathsf{TR}}}[L(\mathcal{Q}|\mathcal{S};\theta)] \right] p^{\mathsf{TR}}(\mathcal{T}_t^{\mathsf{TR}}) \, dt \\
&= \int \left[ \mathbb{E}_{(\mathcal{S},\mathcal{Q})\sim\mathcal{D}_t^{\mathsf{TR}}}[L(\mathcal{Q}|\mathcal{S};\theta)] \right] p^{\mathsf{TE}}(\mathcal{T}_t^{\mathsf{TR}}) \, dt \\
&= \mathbb{E}_{\mathcal{T}_t^{\mathsf{TE}}\sim p^{\mathsf{TE}}(\mathcal{T})} \left[ \mathbb{E}_{(\mathcal{S},\mathcal{Q})\sim\mathcal{D}_t^{\mathsf{TE}}}[L(\mathcal{Q}|\mathcal{S};\theta)] \right] \\
&= R^{\mathsf{TE}},
\end{aligned}
\tag{6}
$$

given that expected weighted meta-training error $R^{\mathsf{IWML}}$ and expected meta-test error $R^{\mathsf{TE}}$ match, optimizing the parameter $\theta$ by minimizing $R^{\mathsf{IWML}}$ can enhance performance on tasks following the meta-test task distribution.

### 4.3 Importance Estimation

In the proposed method, we use the ratio of the densities of meta-test and meta-training tasks at feature vector distribution of task $\mathcal{T}_t$ to approximate the density ratio of $p^{\mathsf{TE}}(\mathcal{T}_t)$ and $p^{\mathsf{TR}}(\mathcal{T}_t)$:

$$
\frac{p^{\mathsf{TE}}(\mathcal{T}_t)}{p^{\mathsf{TR}}(\mathcal{T}_t)} \approx \frac{p^{\mathsf{TE}}(q_t(\mathbf{x}))}{p^{\mathsf{TR}}(q_t(\mathbf{x}))},
\tag{7}
$$

where $q_t(\mathbf{x})$ represents the feature vector distribution of task $\mathcal{T}_t$. In many meta-learning studies (Yao et al., 2019; Wu et al., 2022), task $\mathcal{T}_t$ is represented by its corresponding labeled dataset $\mathcal{D}_t = \{(\mathbf{x}_{tn}, y_{tn})\}_{n=1}^{N_t}$. In our setting, we only have unlabeled datasets in meta-test tasks obtained from the meta-test task distribution in the meta-training phase, which does not allow us to represent the task by its labeled dataset. Therefore,

we assume that marginal feature vector distribution $q(\mathbf{x})$ carries information about joint distribution $q(\mathbf{x}, y)$. This assumption allows us to only use a set of unlabeled data to gain insight into the task. This supposition aligns with the cluster assumption found in semi-supervised learning (Ouali et al., 2020; Van Engelen & Hoos, 2020). This assumption posits that feature vectors, which are close to each other in the feature space, likely share the same label. Based on this assumption, we can leverage $q(\mathbf{x})$ to gain useful insights into $q(\mathbf{x}, y)$. In Section 5, we validated this assumption through experiments.

We employ the Kernel Density Estimation (KDE) methodology to estimate the density of the feature vector distribution. According to the KDE, the densities of meta-test and meta-training are calculated by:

$$p^{\mathsf{TE}}(q_t(\mathbf{x})) = \frac{1}{Z^{\mathsf{TE}}} \left[ \frac{1}{T^{\mathsf{TE}}} \sum_{t'=1}^{T^{\mathsf{TE}}} \exp\left( \frac{-d(q_t(\mathbf{x}), q_{t'}^{\mathsf{TE}}(\mathbf{x}))}{2h^2} \right) \right], \tag{8}$$

and

$$p^{\mathsf{TR}}(q_t(\mathbf{x})) = \frac{1}{Z^{\mathsf{TR}}} \left[ \frac{1}{T^{\mathsf{TR}}} \sum_{t'=1}^{T^{\mathsf{TR}}} \exp\left( \frac{-d(q_t(\mathbf{x}), q_{t'}^{\mathsf{TR}}(\mathbf{x}))}{2h^2} \right) \right], \tag{9}$$

where $q_{t'}^{\mathsf{TR}}(\mathbf{x})$ represents the feature vector distribution of $t'$-th meta-training task, $q_{t'}^{\mathsf{TE}}(\mathbf{x})$ represents the feature vector distribution of $t'$-th meta-test task, $h^2$ is a smoothing scalar parameter, commonly referred to as the bandwidth, and $d(q_t(\mathbf{x}), q_{t'}^{\mathsf{TE}}(\mathbf{x}))$ represents the distance between $q_t(\mathbf{x})$ and $q_{t'}^{\mathsf{TE}}(\mathbf{x})$. Normalization factors denoted as $Z^{\mathsf{TE}}$ and $Z^{\mathsf{TR}}$, are employed to ensure that the sum of the estimated densities equals one. They are calculated by:

$$Z^{\mathsf{TE}} = \sum_{t=1}^{T} \left[ \frac{1}{T^{\mathsf{TE}}} \sum_{t'=1}^{T^{\mathsf{TE}}} \exp\left( \frac{-d(q_t(\mathbf{x}), q_{t'}^{\mathsf{TE}}(\mathbf{x}))}{2h^2} \right) \right], \tag{10}$$

and

$$Z^{\mathsf{TR}} = \sum_{t=1}^{T} \left[ \frac{1}{T^{\mathsf{TR}}} \sum_{t'=1}^{T^{\mathsf{TR}}} \exp\left( \frac{-d(q_t(\mathbf{x}), q_{t'}^{\mathsf{TR}}(\mathbf{x}))}{2h^2} \right) \right], \tag{11}$$

where $q_t(\mathbf{x}) = q_t^{\mathsf{TR}}(\mathbf{x})$ for $t = 1, \ldots, T^{\mathsf{TR}}$, $q_t(\mathbf{x}) = q_{t-T^{\mathsf{TR}}}^{\mathsf{TE}}(\mathbf{x})$ for $t = T^{\mathsf{TR}}, \ldots, T$, total number of meta-training and meta-test tasks represented by $T = T^{\mathsf{TE}} + T^{\mathsf{TR}}$.

To estimate the distance between two distributions, we use Maximum Mean Discrepancy (MMD) (Borgwardt et al., 2006; Gretton et al., 2012). MMD is a distance measure of two probability distributions or two sets of points. Given two sets of feature vector $\mathcal{U}_t = \{\mathbf{x}_{tn}\}_{n=1}^{N_t}$ drawn from $q_t(\mathbf{x})$, $\mathcal{U}_{t'}^{\mathsf{TE}} = \{\mathbf{x}_{t'n}^{\mathsf{TE}}\}_{n=1}^{N_{t'}^{\mathsf{TE}}}$ drawn from $q_{t'}^{\mathsf{TE}}(\mathbf{x})$, empirical estimation of the MMD between these distributions is calculated by:

$$d(q_t(\mathbf{x}), q_{t'}^{\mathsf{TE}}(\mathbf{x})) = \mathrm{MMD}^2(\mathcal{U}_t, \mathcal{U}_{t'}^{\mathsf{TE}}), \tag{12}$$

where,

$$
\begin{aligned}
\mathrm{MMD}^2(\mathcal{U}_t, \mathcal{U}_{t'}^{\mathsf{TE}}) = {} & \frac{1}{N(N-1)} \sum_{n=1}^{N} \sum_{\substack{n \neq n'}}^{N} k(\mathbf{x}_n, \mathbf{x}_{n'}) - \frac{2}{N N_t^{\mathsf{TE}}} \sum_{n=1}^{N} \sum_{n'=1}^{N_t^{\mathsf{TE}}} k(\mathbf{x}_n, \mathbf{x}_{tn'}^{\mathsf{TE}}) \\
& + \frac{1}{N_{t'}^{\mathsf{TE}}(N_{t'}^{\mathsf{TE}} - 1)} \sum_{n=1}^{N_{t'}^{\mathsf{TE}}} \sum_{\substack{n \neq n'}}^{N_{t'}^{\mathsf{TE}}} k(\mathbf{x}_{tn}^{\mathsf{TE}}, \mathbf{x}_{tn'}^{\mathsf{TE}}),
\end{aligned}
\tag{13}
$$

where $k$ is a kernel. Considering that $\mathcal{U}_t$ and $\mathcal{U}_{t'}^{\mathsf{TE}}$ are sets of high-dimensional feature vectors, such as a collection of images, we can use the deep kernel (Wilson et al., 2016) for the calculating MMD. Utilizing deep kernels allows for the acquisition of representations particularly tailored to the given data, which can

potentially enhance the performance of KDE. Therefore, we employ function $g_\phi$ for feature extraction in deep Gaussian kernel calculation, where function $g_\phi : \mathbb{R}^D \to \mathbb{R}^M$ with parameters $\phi$ serves as an encoder for feature extraction, Here, $M$ is the dimension of the feature vector after extraction. The deep Gaussian kernel used in Eq. 13 is given by

$$k_\phi(\mathbf{x}_n, \mathbf{x}_{n'}) = \exp\left(\frac{-\|g_\phi(\mathbf{x}_n) - g_\phi(\mathbf{x}_{n'})\|^2}{2\sigma^2}\right), \tag{14}$$

where $\sigma^2$ is a smoothing scalar parameter, commonly referred to as the bandwidth. Similar to Eq. 12, the distance between $q_t(\mathbf{x})$ and $q_{t'}^{\mathsf{TR}}(\mathbf{x})$ used in Eq. 9 is calculated by:

$$d(q_t(\mathbf{x}), q_{t'}^{\mathsf{TR}}(\mathbf{x})) = \mathrm{MMD}^2(\mathcal{U}_t, \mathcal{U}_{t'}^{\mathsf{TR}}), \tag{15}$$

where $\mathcal{U}_{t'}^{\mathsf{TR}} = \{\mathbf{x}_{tn}^{\mathsf{TR}}\}_{n=1}^{N_t^{\mathsf{TR}}}$ drawn from $q_{t'}^{\mathsf{TR}}(\mathbf{x})$.

Through the importance estimation process, the weight $\mathbf{w}_t$ of meta-training task $\mathcal{T}_t^{\mathsf{TR}}$ is calculated by Eq. 5. The meta-test and meta-training densities of meta-training task in Eq. 5 are estimated using Eq. 8 and Eq. 9, based on feature vectors in $\{\mathcal{D}_t^{\mathsf{TR}}\}_{t=1}^{T^{\mathsf{TR}}}$ and $\{\mathcal{U}_t^{\mathsf{TE}}\}_{t=1}^{T^{\mathsf{TE}}}$. The importance estimation procedures of IWML are shown in Algorithm 1.

### 4.4 Meta-training procedure

We estimate model parameters $\theta$ and encoder parameters $\phi$ in two steps.

- Step 1: We estimate parameters $\phi$ of encoder $g$ for kernel $k$ in Eq. 14. In our experiments, we use prototypical networks (Snell et al., 2017) to train parameter $\phi$ on meta-training tasks $\{\mathcal{T}_t^{\mathsf{TR}}\}_{t=1}^{T^{\mathsf{TR}}}$ by minimizing the expected (unweighted) meta-training loss in Eq. 1 on meta-training datasets $\{\mathcal{D}_t^{\mathsf{TR}}\}_{t=1}^{T^{\mathsf{TR}}}$.

- Step 2: We estimate model parameter $\theta$ by minimizing expected weighted meta-training loss $R^{\mathsf{IWML}}$ in Eq. 4 on meta-training datasets $\{\mathcal{D}_t^{\mathsf{TR}}\}_{t=1}^{T^{\mathsf{TR}}}$. The weight of $t$-th meta-training task $\mathbf{w}_t$ in Eq. 4 is calculated by Algorithm 1 using $t$-th meta-training dataset and unlabeled meta-test datasets $\{\mathcal{U}_t^{\mathsf{TE}}\}_{t=1}^{T^{\mathsf{TE}}}$.

In Step 2, prototypical networks (Snell et al., 2017) and model-agnostic meta-learning (Finn et al., 2017) are used to train parameters $\theta$ in our experiments. The parameters $\theta$ can be trained with various meta-learning methods, such as Meta-SGD (Li et al., 2017), and neural processes (Garnelo et al., 2018). Similarly to $\theta$, parameters $\phi$ of encoder $g$ also can be trained with various methods. The meta-training procedures of IWML are shown in Algorithm 2.

## 5 Experimental Evaluation

### 5.1 Datasets

We evaluated our proposed method using three datasets: miniImageNet, Omniglot, and tieredImageNet. The miniImageNet dataset, first introduced by Vinyals (Vinyals et al., 2016), is a subset of the larger ILSVRC-12 dataset (Russakovsky et al., 2015). It consists of 60,000 color images, each 84×84 pixels in resolution, divided into 100 classes. Each class contains 600 examples.

Omniglot dataset (Lake et al., 2011) includes 1,623 unique, hand-drawn characters from 50 different alphabets. Each character is associated with 20 images, each 28×28 pixels in resolution, all of which were produced by unique human subjects.

The tieredImageNet dataset, initially proposed by (Ren et al., 2018). It comprises 779,165 color images in 608 classes, grouped into 34 higher-level nodes within the ImageNet human-curated hierarchy. This set of nodes is further divided into 20, 6, and 8 disjoint sets for training, validation, and test, respectively. Ren et

---

**Algorithm 1** Importance estimation procedure of IWML

---

**Input**: Meta-training datasets $\{\mathcal{D}_t^{\mathsf{TR}}\}_{t=1}^{T^{\mathsf{TR}}}$, unlabeled meta-test datasets $\{\mathcal{U}_t^{\mathsf{TE}}\}_{t=1}^{T^{\mathsf{TE}}}$, encoder $f$.
**Output**: Weight $\mathbf{w}_t$ for $t$-th meta-training task

1: **for** $i \in T^{\mathsf{TE}}$ **do**
2:  Calculate distance between the $t$-th meta-training task and $t'$-th meta-test task by Eq. 13 using the feature vectors in $\mathcal{D}_t^{\mathsf{TR}}$ and $\mathcal{U}_i^{\mathsf{TE}}$
3: **end for**
4: **for** $j \in T^{\mathsf{TR}}$ **do**
5:  Calculate distance between the $t$-th meta-training task and $t'$-th meta-training task by Eq. 13 using the feature vectors in $\mathcal{D}_t^{\mathsf{TR}}$ and $\mathcal{D}_j^{\mathsf{TR}}$
6: **end for**
7: Calculate normalization factors $Z^{\mathsf{TE}}$ and $Z^{\mathsf{TR}}$ by Eq. 10 and Eq. 11
8: Estimate $p^{\mathsf{TE}}(\mathcal{T}_t^{\mathsf{TR}})$ and $p^{\mathsf{TR}}(\mathcal{T}_t^{\mathsf{TR}})$ by Eq. 8 and Eq. 9
9: Calculate weight $\mathbf{w}_t = p^{\mathsf{TE}}(\mathcal{T}_t^{\mathsf{TR}})/p^{\mathsf{TR}}(\mathcal{T}_t^{\mathsf{TR}})$

---

**Algorithm 2** Meta-training procedure of IWML: RandomSample$(\mathcal{S}, N)$ generates a set of $N$ elements chosen uniformly at random from set $\mathcal{S}$ without replacement.

---

**Input**: Meta-training datasets $\{\mathcal{D}_t^{\mathsf{TR}}\}_{t=1}^{T^{\mathsf{TR}}}$, unlabeled meta-test datasets $\{\mathcal{U}_t^{\mathsf{TE}}\}_{t=1}^{T^{\mathsf{TE}}}$, number of support instances $N_{\mathsf{S}}$, number of query instances $N_{\mathsf{Q}}$, batch size $B$.
**Output**: Model parameters $\theta$

1: randomly initialize $\theta$, initialize loss $J \leftarrow 0$
2: **while** not done **do**
3:  Select task indices for a mini-batch, $\mathcal{M} \leftarrow$ RandomSample($\{1, \cdots, T^{\mathsf{TR}}\}$, $B$)
4:  **for** $t \in \mathcal{M}$ **do**
5:   Generate support set, $\mathcal{S} \leftarrow$ RandomSample($\mathcal{D}_t^{\mathsf{TR}}$, $N_{\mathsf{S}}$)
6:   Generate query set, $\mathcal{Q} \leftarrow$ RandomSample($\mathcal{D}_t^{\mathsf{TR}} \setminus \mathcal{S}$, $N_{\mathsf{Q}}$)
7:   Calculate weight $\mathbf{w}_t$ by Algorithm 1 using $\mathcal{D}_t^{\mathsf{TR}}$ and $\{\mathcal{U}_t^{\mathsf{TE}}\}_{t=1}^{T^{\mathsf{TE}}}$
8:   Calculate weighted loss by Eq. 2 and $\mathbf{w}_t$ , $J \leftarrow J + \mathbf{w}_t L(\mathcal{Q}|\mathcal{S}; \theta)$, and its gradients
9:  **end for**
10:  Update model parameters $\theta$ using loss $J/B$ and its gradient
11: **end while**

---

al. (2018) contend that splitting closer to the root of the ImageNet hierarchy leads to a more challenging but more realistic scenario, where test classes are less similar to training classes.

For both miniImageNet and Omniglot, we simulated a task shift scenario by splitting all classes within each dataset into two clusters based on MMD distance. Specifically, first, we randomly selected two classes. Next, for each class, we calculated its MMD distance to these two selected classes using Eq. 13, clustering each class to the one it was closer to. To ensure an adequate number of classes in each cluster, we re-split the miniImageNet datasets if a cluster contained fewer than 15 classes. For Omniglot, we mandated that a cluster must have no fewer than 50 classes. Let denote the two clusters A and B. The meta-training-validation datasets consisted of all classes within cluster A and randomly selected 10% of the classes within cluster B. The meta-training datasets comprised 70% of the classes from the meta-training-validation datasets, also selected without replacement. The meta-validation datasets consisted of the remaining 30% of the classes from the meta-training-validation datasets that were not selected for meta-training. The meta-test datasets was composed of 45% of the classes randomly selected from cluster B not used for meta-training-validation. The unlabeled datasets for importance estimation was made up of the remaining 45% of the classes from cluster B not used for meta-training-validation or meta-test. To ensure overlap between the meta-training and meta-test task distributions, the meta-training-validation datasets included all classes from cluster A, as well as 10% of the classes from cluster B. For tieredImageNet, because the test classes are less similar to the training classes, we followed the original paper's (Ren et al., 2018) method to split the training, validation, and test datasets. The unlabeled datasets used for importance estimation consisted of 40 classes

randomly selected from the test dataset without replacement. The remaining classes in the test dataset, which were not used for importance estimation, formed the new test dataset. The meta-training datasets comprised all the classes from the training dataset, the meta-validation datasets included all the classes from the validation dataset, and the meta-test datasets contained all the classes from the new test dataset. Since the classes in the meta-test and importance estimation datasets did not overlap, the datasets used for computing importance weights during the meta-training phase were entirely separate from those used for the meta-test.

## 5.2 Comparative Methods

We compared the proposed method with the following methods: (1) Model-agnostic Meta-Learning (MAML) (Finn et al., 2017), which is a gradient-based meta-learning algorithm. (2) Prototypical Networks (PN) (Snell et al., 2017), which is a metric-based algorithm. (3) Meta-Mix (Yao et al., 2021a) and Adversarial Task Up-sampling (ATU) (Wu et al., 2022), which are state-of-the-art task augmentation strategies for meta-learning, as we introduced in Section 2.5. and (4) Weighted Meta-Learning (WML) (Cai et al., 2020), which is a method aimed at addressing task shift, as we introduced in Section 2.4. While MAML, PN, Meta-Mix, and ATU leverage labeled meta-training datasets during their meta-training phase, WML and our method utilize both labeled meta-training datasets and unlabeled datasets in tasks obtained from the meta-test task distribution. To mimic real-world situations more closely, We use unlabeled datasets from tasks that differ from the meta-test task.

## 5.3 Experimental Settings

Our network architecture, closely mirroring the one used in the referenced study (Vinyals et al., 2016), comprises four convolutional blocks. Each block consists of a $3 \times 3$ convolution with 128 filters, a subsequent batch normalization layer (Ioffe & Szegedy, 2015), a ReLU nonlinearity, and finally a $2 \times 2$ max-pooling layer. When applied to $28 \times 28$ Omniglot images and $84 \times 84$ miniImageNet images, this architecture produces a 64-dimensional and 1600-dimensional output space, respectively. MAML, Meta-Mix, and ATU all necessitate a classifier at the final layer. In alignment with the baseline classifier utilized in the study (Vinyals et al., 2016), we feed the output of the last layer into the soft-max function. The loss function is defined as the cross-entropy error between the predicted and the actual label. For PN, we employ the Euclidean distance to compute the distance metric. We carry out a single gradient update with a fixed step size $\alpha = 0.05$ in the inner loop, while Adam (Kingma & Ba, 2014) serves as the outer loop optimizer with step size $\beta = 10^{-4}$. Our model and WML were trained using PN, Meta-Mix, and ATU were trained using MAML. Due to the original WML being limited to regression scenarios and constrained to the use of squared and hinge loss as its loss functions, in our experiments, to adapt WML for classification scenarios, we use Eq. 12 to calculate the distance between tasks.

During the meta-training phase, we trained the proposed IWML and the comparative methods using 250,000 meta-training tasks with a meta-batch size of 50. Since both WML and the proposed IWML require unlabeled data to compute weights, we used 30 importance estimation tasks, which consist only of unlabeled data, for weight estimation. However, the other comparative methods did not use any additional unlabeled data during the meta-training phase. We employed 30 validation tasks for early stopping. In the meta-test phase, we computed the classification accuracy based on an average of over 30 meta-test tasks. Meta-training, validation, meta-test, and importance estimation tasks were randomly generated from their respective datasets, each task was constructed using a 5-way setting with 1-shot, 2-shot, or 3-shot configurations for miniImageNet and tieredImageNet, and 20-way with the same shot variations for the Omniglot datasets. We applied the same settings as used in training the PN on each dataset for training encoder $g_\phi$. For smoothing scalar parameters $\sigma$ and $h$ used in Eq. 14 and Eq. 9, we designate both as task-specific smoothing scalar parameters. For $\sigma$, its value for each task is determined by calculating the median of $\{\|g_\phi(\mathbf{x}_n) - g_\phi(\mathbf{x}_{n'})\|^2\}_{n'=1}^{N}$. Similarly, for $h$, it is determined using the median of $\{d(q(\mathbf{x}), q_{t'}^{\mathsf{TR}}(\mathbf{x}))\}_{t'=1}^{T^{\mathsf{TR}}}$. Additionally, $h$ in Eq. 8, Eq. 10, and Eq. 11 was determined in the similar calculation with Eq. 9.

| Method | 1-shot | 2-shot | 3-shot |
|--------|--------|--------|--------|
| MAML | 30.4±1.8 | 32.1±2.5 | 33.7±3.8 |
| PN | 31.0±3.0 | 34.1±2.7 | 36.5±3.4 |
| Meta-Mix | 29.8±2.9 | 34.8±2.8 | 34.4±3.7 |
| ATU | 28.8±3.6 | 35.0±2.3 | 34.8±5.4 |
| WML | 31.7±2.4 | 35.9±2.9 | 37.8±3.6 |
| Ours | **34.4±3.1** | **38.0±3.7** | **43.6±3.0** |

Table 1: Averaged accuracies (%) and standard errors are presented for 5-way classification on meta-test tasks using miniImageNet. Values in bold typeface are not statistically different from the best-performing method at the 5% level by a paired t-test.

| Method | 1-shot | 2-shot | 3-shot |
|--------|--------|--------|--------|
| MAML | 72.1±0.6 | 82.6±0.7 | 85.6±1.0 |
| PN | 68.9±0.7 | 83.7±0.4 | 87.4±0.9 |
| Meta-Mix | 71.6±1.5 | 82.4±1.1 | 86.3±1.3 |
| ATU | 72.7±1.3 | 83.9±0.6 | 85.2±0.9 |
| WML | 69.3±1.7 | 81.9±0.9 | 88.4±0.7 |
| Ours | **92.9±0.7** | **96.4±0.4** | **97.1±0.7** |

Table 2: Averaged accuracies (%) and standard errors are presented for 20-way classification on meta-test tasks using Omniglot. Values in bold typeface are not statistically different from the best-performing method at the 5% level by a paired t-test.

| Method | 1-shot | 2-shot | 3-shot |
|--------|--------|--------|--------|
| MAML | 31.3±3.6 | 32.9±3.1 | 35.8±2.9 |
| PN | 32.9±1.5 | 37.1±1.0 | 44.1±1.4 |
| Meta-Mix | 32.2±3.8 | 34.0±2.2 | 37.7±4.0 |
| ATU | 32.8±3.1 | 36.7±5.5 | 37.8±4.1 |
| WML | 33.6±1.2 | 38.7±1.8 | 45.6±2.0 |
| Ours | **38.3±2.3** | **44.6±2.1** | **51.8±3.1** |

Table 3: Averaged accuracies (%) and standard errors are presented for 5-way classification on meta-test tasks using tieredImageNet. Values in bold typeface are not statistically different from the best-performing method at the 5% level by a paired t-test.

### 5.4 Results

We show the performance of the three datasets in Tables 1, 2, and 3. On all three datasets, the proposed method consistently outperforms the baseline methods. WML outperforms MAML and PN but is inferior to the proposed method. One reason for this is that WML faces difficulty in computing more accurate weights for each meta-training task, especially when using tasks that align with the meta-test task distribution but will not appear in the meta-test phase. In contrast, our method calculates weights by estimating the density of each meta-training task within the distributions of both meta-test and meta-training tasks. This enables our method to compute accurate weights, even when utilizing tasks that conform to the meta-test task distribution but are absent from the meta-test phase. The performance of Meta-Mix and ATU does not improve over MAML. One reason is that, in the case of task shift, due to the different distributions between meta-training and meta-test tasks, they fail to generate tasks similar to the meta-test tasks using the given meta-training tasks. However, the proposed method enables the model to leverage additional unlabeled data from the meta-test task distribution, allowing it to focus more on meta-training tasks that closely align with the meta-test tasks during the meta-training process, resulting in better performance. Table 4 shows the training time for PN, WML, and the proposed method. Although our method takes longer than PN, it is much shorter than that of WML. This is because WML assumes that there is only one meta-test task during the meta-test phase. When there are multiple meta-test tasks during the meta-test phase, WML needs to train specific model parameters for each meta-test task.

| PN | WML | Ours |
|----|-----|------|
| 0.2 | 6.7 | 0.3 |

Table 4: Training computational time in hours under the 1-shot 5-way setting on the miniImageNet.

For the ablation study, we evaluated the performance without using MMD or the deep kernel in Eq. 8 and Eq. 9. In the experiments without MMD, the distance between feature vector distributions was calculated

as follows:

$$d(q_t(\mathbf{x}), q_{t'}^{\mathsf{TE}}(\mathbf{x})) = \frac{1}{N N_t^{\mathsf{TE}}} \sum_{n=1}^{N} \sum_{n'=1}^{N_t^{\mathsf{TE}}} ||g_\phi(\mathbf{x}_n) - g_\phi(\mathbf{x}_{tn'}^{\mathsf{TE}})||^2. \tag{16}$$

For experiments without the deep kernel, the Gaussian kernel used in Eq.13 is given by:

$$k(\mathbf{x}_n, \mathbf{x}_{n'}) = \exp\left(\frac{-||\mathbf{x}_n - \mathbf{x}_{n'}||^2}{2\sigma^2}\right). \tag{17}$$

The results in Table 5 indicate a performance decline for the proposed method when MMD is not used. This is because MMD can more accurately reflect the distribution differences between two sets of data compared to the mean of point-to-point distances. Similarly, performance also decreases when the deep kernel is not used. This is because deep kernels can obtain feature representations specifically suited to the given data through an encoder, potentially enhancing the performance of KDE compared to general kernel functions.

| Ours | Without MMD | Without deep kernel |
|------|-------------|---------------------|
| 34.4± 3.1 | 32.5±2.8 | 32.9±2.7 |

Table 5: Ablation study: The averaged accuracies (%) and standard errors on the meta-test tasks under the 1-shot 5-way setting on the miniImageNet dataset were obtained using the proposed method with different MMD and deep kernel choices of Eq. 8 and Eq. 9.

For the encoder $g_\phi$, we evaluated the performance when trained using only labeled meta-training data and when trained using joint labeled meta-training data and unlabeled data obtained from the meta-test task distribution. The results are presented in Table 6, where "ours" indicates the performance using only labeled data, and "ours with autoencoder" indicates the performance using both labeled and unlabeled data. In the experiment using both labeled and unlabeled data, we considered $g_\phi$ as part of an autoencoder (Rumelhart et al., 1986). The training process was divided into two stages: (1) When training with labeled meta-training data, a PN was added and trained together with the autoencoder. The autoencoder was trained by minimizing both the error of the PN and the reconstruction error. (2) When training with unlabeled data, the autoencoder was trained by minimizing the reconstruction error alone.

The results in Table 6 indicate that training the encoder $g_\phi$ using joint labeled meta-training data and unlabeled data obtained from the meta-test task distribution achieves better performance compared to training with only labeled meta-training data. This is because when the encoder is trained solely with labeled meta-training data, it struggles to derive suitable representations for data from the meta-test task distribution due to task shift, leading to decreased performance. In contrast, training $g_\phi$ with both labeled meta-training data and unlabeled data from the meta-test task distribution allows the encoder to more effectively obtain suitable representations for data from both task distributions, thereby enhancing overall performance.

| Ours | Ours with autoencoder |
|------|------------------------|
| 34.4± 3.1 | 35.6± 2.5 |

Table 6: The averaged accuracies (%) and standard errors on the meta-test tasks under the 1-shot 5-way setting on the miniImageNet dataset where encoder $g_\phi$ as part of an autoencoder are trained using joint labeled meta-training data and unlabeled data obtained from the meta-test task distribution.

### 5.5 Visualization of the task shift

We visualize the task shift in the miniImageNet dataset under a 5-way 1-shot setup. Specifically, we randomly selected 50 meta-training tasks from the meta-training datasets and 50 meta-test tasks from the meta-test datasets. We used t-SNE (Van der Maaten & Hinton, 2008) with the MMD distance for visualization. The results shown in Figure 2 reveal that although there is a significant difference between the distribution of meta-training tasks and meta-test tasks, a portion of the meta-training tasks are closer to the meta-test

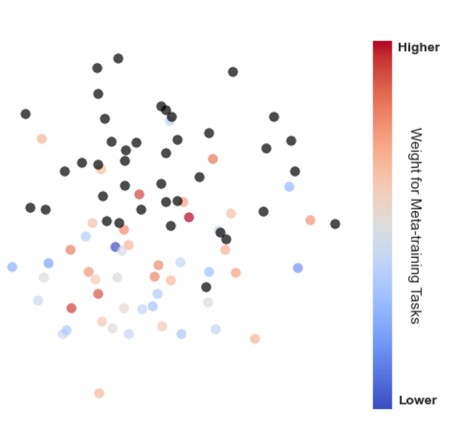

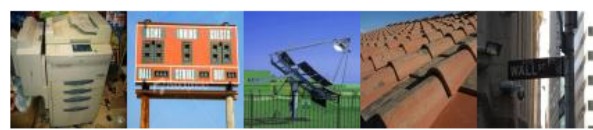

(a) Meta-training tasks with weights: 1.324

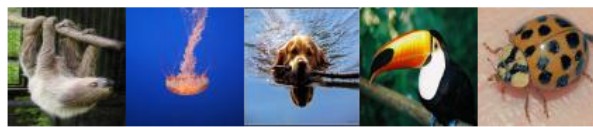

(b) Meta-training tasks with weights: 0.816

(c) Meta-test task

Figure 2: Visualizing the task shift scenario in mini-ImageNet: Each black point represents a meta-test task, while points in other colors indicate meta-training tasks. Tasks with higher assigned weights are depicted in richer shades of red.

Figure 3: Examples of images in meta-training tasks with weights 1.324/0.816 are represented by (a) and (b), while those in a meta-test task are represented by (c). These tasks are generated from miniImageNet datasets using a 1-shot, 5-way setting.

tasks. The proposed method can assign higher weights to these meta-training tasks that are closer to the meta-test tasks.

Figure 3 shows examples of images in meta-training tasks with weights 1.324/0.816 and those in a meta-test task. As depicted in Figure 3(c), the data in meta-test tasks consists of images related to animals and insects. The proposed method assigned a high weight to a task depicted in Figure 3(a) that contains more animal and insect images. In contrast, tasks with fewer animal images are assigned lower weights as shown in Figure 3(b). This indicates that our method enables us to find meta-training tasks that closely resemble the meta-test tasks. By assigning high weights to these tasks, we can ensure that the distribution of meta-training tasks aligns as closely as possible with that of the meta-test tasks.

## 6 Conclusion

We proposed a meta-learning method that improves performance under task shift, where the task distributions differ between the meta-training and meta-test phases. In the proposed method, weights are assigned to tasks using the density ratio between these distributions. Experiments demonstrate that our method outperforms existing approaches. However, the proposed method has its limitations. Specifically, (1) Our evaluation focuses on the miniImageNet, Omniglot, and tieredImageNet datasets, which, although widely used, do not capture the full range of challenges posed by more complex benchmarks such as Meta-Dataset (Triantafillou et al., 2019), VTAB (Zhai et al., 1910), and WILDS (Sagawa et al., 2021). Future work will aim to address this by applying the method to these benchmarks and assessing its robustness in more challenging scenarios. (2) When the accessible unlabeled dataset and the meta-test dataset follow different distributions, the performance of our proposed method will decrease. (3) when the density of meta-training tasks used as the denominator in the density ratio nears zero, there is a risk that the calculation of weights becomes unfeasible. Therefore, it is preferable to estimate weights directly without estimating densities. For example, methods proposed to directly estimate weights to address covariate shift (Sugiyama et al., 2008; Bickel et al., 2009). In future research, we plan to extend the proposed method to estimate weights directly without needing to estimate the density of tasks.

## Broader Impact Statement

The proposed method uses unlabeled datasets derived from tasks that adhere to the meta-test task distribution to compute weights for each meta-training task. Consequently, there is a potential risk that users might use biased datasets to calculate these weights without careful consideration, potentially leading to biased predictions. We encourage research into developing methods for automatically detecting biased datasets.

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
