# OpenReview forum: "Meta-Learning under Task Shift"
_TMLR — Accepted by TMLR_

### Review · Reviewer_yGSH · 2024-06-06

**Summary Of Contributions:**

This paper addresses the meta-learning method in the presence of distribution shifts. Weights for meta-training tasks are estimated using Kernel Density Estimation, and an importance-weighted meta-learning loss is utilized. Experiments on several datasets demonstrate that the proposed weighted meta-loss outperforms previous related methods.

**Audience:**

Yes

**Broader Impact Concerns:**

No.

**Claims And Evidence:**

Yes

**Requested Changes:**

1. Clarify what the new technique proposed in this paper is.
2. Provide more analysis on using unlabeled datasets in the presence of distribution shifts.
3. Present detailed information with clear task descriptions for the experiments.

**Strengths And Weaknesses:**

## Strengths
1. The presentation of this paper is easy to follow and well-organized.
2. The motivation behind the proposed weighted meta-loss is clear and reasonable.
3. The method for estimating importance weights is logically sound.
4. The experimental results show significant improvements over previous methods.

## Weaknesses
1. It is unclear whether the Kernel Density Estimation is an existing technique or a new one proposed in this paper.
2. The paper focuses on estimating weights for each meta-training task, but it does not address the scenario of having only a single meta-training task.
3. The paper assumes access to an unlabeled meta-test dataset. However, it does not consider the situation where there is a significant shift between the accessible unlabeled dataset and the meta-test dataset.
4. The experimental settings regarding the datasets and task descriptions are not clearly detailed. For example, it is unclear which parts are meta-training tasks and which are meta-test tasks. More importantly, detailed information about the unlabeled dataset used is lacking.

---

> ### Author Response · Authors · 2024-07-05
>
> Dear reviewer
>
> Thank you very much for your insightful comments and valuable feedback. We appreciate the time and effort you have taken to review our manuscript and provide suggestions for improvement.
>
> ------
>
> Reviewer Comment:  It is unclear whether the Kernel Density Estimation is an existing technique or a new one proposed in this paper.
>
> Response:  We would like to clarify that both Kernel Density Estimation (KDE) and Maximum Mean Discrepancy (MMD) are existing, well-established methods. However, using KDE and MMD to estimate task distribution is a new method proposed in this paper. Additionally, employing MMD within KDE is also a new technique introduced in this work. These new methods allow us to estimate the importance weights for meta-learning tasks, effectively accounting for task shifts.
>
> Manuscript Revisions:  To improve clarity and address your concern, we have updated Section 1 of the revised manuscript to explicitly state that using KDE and MMD to estimate task distribution is a new method proposed in this paper. Additionally, employing MMD within KDE is also a new technique introduced in this work.
>
> -------
>
>
> Reviewer Comment:  The paper focuses on estimating weights for each meta-training task, but it does not address the scenario of having only a single meta-training task.
>
> Response:  We acknowledge that our original manuscript did not explicitly address the scenario of having only a single meta-training task. However, in many meta-learning studies, as introduced in Section 2.3, it is typical to consider multiple meta-training tasks during the meta-training phase. Training on multiple diverse tasks allows the model to learn general features and patterns that can be transferred to new tasks. In contrast, training on a single task limits this ability, making it harder for the model to adapt to new tasks.
>
> ------
>
> Reviewer Comment:  The paper assumes access to an unlabeled meta-test dataset. However, it does not consider the situation where there is a significant shift between the accessible unlabeled dataset and the meta-test dataset.
>
> Response:  The proposed method assumes that the accessible unlabeled dataset and the meta-test dataset follow the same distribution. If there is a significant shift between the accessible unlabeled dataset and the meta-test dataset, the assumption will no longer hold, resulting in a decline in performance. However, in many practical applications, the accessible unlabeled dataset is generally collected from the same distribution as the meta-test tasks. For example, in image recognition tasks, if the meta-test tasks involve classifying microscope images, researchers typically collect microscope images to calculate importance weights, rather than data that is significantly different from the meta-test distribution. Additionally, in research related to covariate shift and unsupervised domain adaptation, as introduced in Sections 2.1 and 2.2, it is typically assumed that the accessible unlabeled dataset and the test dataset do not differ significantly.
>
> Manuscript Revisions:  To improve clarity and address your concern, we have updated Section 6 of the revised manuscript to explicitly state that when the accessible unlabeled dataset and the meta-test dataset follow different distributions, the performance of our proposed method will decrease.
>
> -------

---

> > ### Author Response · Authors · 2024-07-05
> >
> > Reviewer Comment:  The experimental settings regarding the datasets and task descriptions are not clearly detailed. For example, it is unclear which parts are meta-training tasks and which are meta-test tasks. More importantly, detailed information about the unlabeled dataset used is lacking.
> >
> > Response:  We acknowledge that our original explanation on Section 5.1, paragraph 4, might not have been clear enough. Here, we provide a more detailed clarification. For both miniImageNet and Omniglot, we simulated a task shift scenario by splitting all classes within each dataset into two clusters based on MMD distance. Specifically, we randomly pick two classes. For each class, we calculate its MMD distance to these two selected classes using Eq.13, clustering each class around the one it is closer to. To ensure an adequate number of classes in each cluster, we re-split the miniImageNet datasets if a cluster contains fewer than 15 classes. For Omniglot, we mandate that a cluster must have no fewer than 50 classes. The two clusters are A and B. The meta-training dataset consists of 70\% of the classes randomly selected from cluster A and 10\% of the classes randomly selected from cluster B to ensure some overlap between the meta-training and meta-test task distributions. The meta-validation dataset consists of the remaining 30\% of the classes from cluster A. The meta-test dataset is formed from 45\% of the classes randomly selected from cluster B. The unlabeled dataset for importance estimation is composed of the remaining 45\% of the classes from cluster B. For tieredImageNet, because the test classes are less similar to the training classes, we followed the original paper's[R1] method to split the training, validation, and test datasets, and randomly selected 40 classes from the test dataset for importance estimation without labels. Meta-training, validation, meta-test, and importance estimation tasks were randomly generated from their respective datasets, with importance estimation tasks only having unlabeled data.
> >
> > Manuscript Revisions:  To improve clarity and address your concerns, we have updated Section 5.1, paragraph 4 of the revised manuscript to explicitly explain how the training, validation, and test datasets were split for each dataset in our experiments. Additionally, we clarified that the meta-training and meta-test tasks were randomly generated from the training and test datasets, respectively. Furthermore, we clearly stated that the unlabeled data used for weight computation was randomly selected from the test dataset.
> >
> > [R1] Meta-learning for semi-supervised few-shot classification, arXiv 2018.
> >
> > ------
> >
> > Thank you once again for your valuable feedback, which has helped us improve the clarity and completeness of our manuscript.
> >
> >
> > Best regards

---

### Review · Reviewer_PArP · 2024-06-08

**Summary Of Contributions:**

This paper delves into the issue of task shifting within meta-learning, specifically addressing the discrepancy between the distributions of meta-training and meta-testing datasets. It proposes a solution to assign importance weights to various meta-training tasks, utilizing kernel density estimation (KDE) and Maximum Mean Discrepancy (MMD) distance.

**Audience:**

Yes

**Broader Impact Concerns:**

Upon reviewing the work, I did not find any significant ethical concerns.

**Claims And Evidence:**

Yes

**Requested Changes:**

**Requested Changes that are critical to securing my recommendation for acceptance:**
- The Weakness 1.  Addressing this issue is crucial in my view, as it will substantiate the effectiveness of the proposed method. If this concern is satisfactorily resolved, I am inclined to recommend acceptance of the work.

**Requested Changes that can strengthen the work**
- The Weakness 2 & 3.

**Strengths And Weaknesses:**

**Strengths**
- This paper is well-structured and easy to comprehend.
- The motivation to deal with the task shifts problem within meta-learning is clear， and the proposed reweighting methods align with intuitive understanding.
- The inclusion of visualizations (Fig.2 and Fig.3) to showcase the learned weights for meta-training tasks is appreciated, as it validates the effectiveness of the proposed reweighting methods.

**Weaknesses**
- The construction of the task shift dataset also relies on Equation (13). How does the model perform in other task shift settings? I am skeptical about the generalization ability of the method in other task shift scenarios because there is a strong relationship between the task construction and the method itself.
- This paper proposes assigning importance weights to different tasks within a meta-training task, which is closely related to [1]. I believe the authors need to draw a clear comparison between these two closely related methods.
- I suggest that the author should offer specific weights within Fig.3 to facilitate a more effective comparison, rather than merely illustrating high and low weights.

[1] Meta-learning with an adaptive task scheduler, NeurIPS 2021.

---

> ### Author Response · Authors · 2024-07-05
>
> Dear reviewer
>
> Thank you very much for your insightful comments and valuable feedback. We appreciate the time and effort you have taken to review our manuscript and provide suggestions for improvement.
>
> ------
>
> Reviewer Comment:  The construction of the task shift dataset also relies on Equation (13). How does the model perform in other task shift settings? I am skeptical about the generalization ability of the method in other task shift scenarios because there is a strong relationship between the task construction and the method itself.
>
> Response:  To address your concerns, we would like to emphasize that the experiments on the tieredImageNet dataset did not simulate task shift using Equation (13). In this paper, we used three datasets for our experiments. As you mentioned, two of these datasets were used to simulate task shift based on MMD. However, for the experiments on the tieredImageNet dataset, we followed the original paper's [R1] method to split the training, validation, and test datasets. In tieredImageNet, this ensures that the training and test classes do not have any semantic overlap, resulting in test classes that are less similar to training classes, making it a more realistic task shift setting. The splitting method for tieredImageNet is discussed on Section 5.1, paragraph 3 of our paper. The experimental results using the tieredImageNet dataset (Table 3) indicate that our method is effective under various task shift settings. Additionally, we have revised the definition of task shift in Section 4.1 for better clarity. Task shift means that the meta-training and meta-test tasks follow different probability distributions, but the supports of the meta-training and meta-test task distributions are overlapped. As shown in Eqs. 6 and 7, our method is effective under various task shift settings as long as the cluster assumption (Section 4.3) is satisfied.
>
> Manuscript Revisions:  To improve clarity and address your concerns, we have updated Section 5.1, paragraph 4 of the revised manuscript to explicitly emphasize that we simulated task shift scenarios using both MMD and tieredImageNet. The results in Table 3 demonstrate that our method is effective under various task shift settings. Additionally, we have revised the definition of task shift in Section 4.1 for better clarity. Under this definition, our method is effective as long as the cluster assumption (Section 4.3) is satisfied.
>
> [R1] Meta-learning for semi-supervised few-shot classification, arXiv 2018.
>
> ------
>
> Reviewer Comment:  This paper proposes assigning importance weights to different tasks within a meta-training task, which is closely related to [1]. I believe the authors need to draw a clear comparison between these two closely related methods.
>
> Response:  We acknowledge the close relationship between our method and the Adaptive Task Scheduler (ATS) proposed in [1]. Here, we provide a detailed comparison. The main difference between our proposed method and ATS lies in the problems they aim to solve. ATS introduces weights to optimize the generalization capacity of the model and to avoid meta-overfitting, whereas our proposed method addresses meta-learning under task shift. These are two distinct problems, and because of the differences in the problems they solve, the methods for calculating weights are also different. ATS calculates weights based on the loss on the query set and the gradient similarity between the support set and the query set. In contrast, our method calculates weights based on the ratio of meta-test and meta-training task densities. Although ATS is effective in optimizing the generalization capacity of the model and avoiding overfitting, it does not address the issue of task shift.
>
>
> Manuscript Revisions: We have added a comparison of the two methods in Section 2.3 of the revised manuscript.
>
> [1] Meta-learning with an adaptive task scheduler, NeurIPS 2021.
>
> ------
>
> Reviewer Comment:  I suggest that the author should offer specific weights within Fig.3 to facilitate a more effective comparison, rather than merely illustrating high and low weights.
>
> Response:  Thank you for your suggestion regarding the presentation of weights in Figure 3. We understand that providing specific weights, rather than just illustrating high and low weights, would facilitate a more effective comparison. In Figure 3(a), the weight of the meta-training task is 1.324, and in Figure 3(b), the weight of the meta-training task is 0.816. We will update Figure 3 in the revised manuscript to include these specific numerical values for better clarity and comparison.
>
> Manuscript Revisions:  We have updated Figure 3 in the revised manuscript to include these specific numerical values for enhanced clarity and comparison.
>
> ------
>
> Thank you once again for your valuable feedback, which has helped us improve the clarity and completeness of our manuscript.
>
> Best regards

---

> > ### Comment · Reviewer_PArP · 2024-07-21
> > **Post-rebuttal comments from reviewer**
> >
> > Thanks for the response. My major concerns have been addressed. Overall I feel positive about the work, and I have updated my score accordingly.

---

### Review · Reviewer_ye1d · 2024-06-21

**Summary Of Contributions:**

This paper introduces a method to perform meta-learning under task shift. Specifically, when the meta-test distribution differs from the meta-training distribution, there is currently no algorithm that ensures that the model learned during meta-train actually performs well during meta-test.
This paper presents an importance weighting based approach to fix this.

**Audience:**

Yes

**Broader Impact Concerns:**

The paper does not have a broader impact statement.
Please make sure to provide this

**Claims And Evidence:**

Yes

**Requested Changes:**

Please see a list of the questions posed above to be addressed as well as the weaknesses section.

**Strengths And Weaknesses:**

## Strengths ##
1. The paper tackles an important problem that is a missing piece of the existing literature
2. The paper is well motivated and written with good use of notation that makes it relatively easy to understand
3. At a high level, the method present is simple (since it makes use of existing pieces) though I do not doubt that fitting everything together was not easy

## Weaknesses ##
1. Some missing ablations.

      a) No ablating MMD^2 or Deep Kernel as choices for use in KDE

      b) $\phi$ is trained on only the meta-train set. Would have been interesting to also see it train on the joint meta-train/unlabelled meta-test data. Especially if it is the case that meta-train and meta-test have different supports.

2.  Need for more robust definition of task shift. The paper current defines task shift as $p^{TE}\left(T_{t} \right) \neq p^{TR}\left(T_{t} \right)$ but there are many ways in which this could manifest -- like non-overlapping input/output supports, covariate shift in the input spaces. **Please make clear what types of shift your method is targetted at -- or if it is supposed to subsume all types of shifts, then show evidence of this**

3. Not enough information about how baselines were implemented. Specifically for the WML method -- do you rerun meta-training  for each individual meta-test task ? My intuition was that WML would perform best (since the weights are learned per meta-test task) but would be too computationally intensive because it's being run for each individual meta-test task. Is your method outperforming WML partially because you aggregate the unlabelled meta-test data and so there is some transfer going on? Can you provide more details about the WML setup ?


## Questions  ##
1. Why just do 1-step GD on the inner loop ? What happens if you do > 1 step in the meta-learning inner loop ?
2. I'm not sure how to phrase this question but it feels a bit circular, if the Dataset you created for experimentation uses the MMD distance to "simulate task shift" when it's the same distance you use in your method construction. Even if the distinction is more nuanced, I think a more "realistic" task shift could be some combination of dataset from the meta-dataset [1] paper.


[1] https://arxiv.org/abs/1903.03096

---

> ### Author Response · Authors · 2024-07-05
>
> Dear reviewer
>
> Thank you very much for your insightful comments and valuable feedback. We appreciate the time and effort you have taken to review our manuscript and provide suggestions for improvement.
>
> ------
>
> Reviewer Comment:
> Some missing ablations.
>
> (a) No ablating MMD or Deep Kernel as choices for use in KDE
>
> (b) $\phi$ is trained on only the meta-train set. Would have been interesting to also see it train on the joint meta-train/unlabelled meta-test data. Especially if it is the case that meta-train and meta-test have different supports.
>
> Response:  Thank you for your suggestion regarding the ablation study. In response to your suggestion (a), we have included a discussion on the ablation study of MMD and the deep kernel in Section 5.4, paragraph 2 of the revised manuscript. The performance results without using MMD or the deep kernel are shown in Table 5. Specifically, in the experiment without MMD, as described in Eq. 16 of the revised version, we used the mean of point-to-point distances instead of the MMD distance. In the experiment without the deep kernel, as described in Eq. 17 of the revised version, we used a Gaussian kernel instead of a deep Gaussian kernel. The results in Table 5 indicate that the performance declines when the proposed method does not use MMD or the deep kernel.
>
> In response to your suggestion (b), we have included a discussion in Section 5.4 of the revised manuscript about training $\phi$ using labeled meta-training data and unlabeled data obtained from the meta-test task distribution. The results are presented in Table 6, where "ours" indicates the performance using only labeled meta-training data, and "ours with autoencoder" indicates the performance using both labeled meta-training data and unlabeled data obtained from the meta-test task distribution. In the experiment using both labeled and unlabeled data, we trained $\phi$ using an autoencoder[R1]. The detailed experimental process is provided in Section 5.1 of the revised manuscript. The results in Table 6 indicate that training the encoder $\phi$ using joint labeled meta-training data and unlabeled data from the meta-test task distribution achieves better performance compared to training with only labeled meta-training data.
>
>
>
> Manuscript Revisions:  In the revised manuscript, we have included Table 5 to show the performance of our proposed method without using MMD or the deep kernel. The specific experimental settings can be found in Section 5.4, paragraph 2 of the revised manuscript. In this section, we also discuss the reasons for the performance degradation when MMD or the deep kernel is not used. Furthermore, we have added Table 6, which shows the performance of $\phi$ when trained on different datasets. The specific experimental settings can be found in Section 5.4, paragraph 3 of the revised manuscript. In paragraph 4, we provide a discussion on the results shown in Table 6.
>
> [R1] Learning Internal Representations by Error Propagation. MIT Press, 1986.
>
> ------
>
> Reviewer Comment:  Need for more robust definition of task shift. The paper current defines task shift as $p^{TE}\left(T_{t} \right) \neq p^{TR}\left(T_{t} \right)$ but there are many ways in which this could manifest -- like non-overlapping input/output supports, covariate shift in the input spaces. \textbf{Please make clear what types of shift your method is targetted at -- or if it is supposed to subsume all types of shifts, then show evidence of this}
>
> Response:  Thank you for your suggestion regarding the task shift definition. In the revised manuscript, we have clarified the definition of task shift in Section 4.1. Task shift means that the meta-training and meta-test tasks follow different probability distributions, but the supports of the meta-training and meta-test task distributions are overlapped. Under this definition, as shown in Eqs. 6 and 7, our method is effective under various task shift settings as long as the cluster assumption (Section 4.3) is satisfied. As you mentioned, when there is no overlap between the support datasets of the meta-test tasks and the meta-training tasks, our method will face difficulties. This is because, in such cases, no meta-training task is relevant for addressing the meta-test tasks.
>
>
> Manuscript Revisions:  We have revised the definition of task shift in Section 4.1 for better clarity. Task shift means that the meta-training and meta-test tasks follow different probability distributions, but the supports of the meta-training and meta-test task distributions are overlapped.
>
> ------

---

> > ### Author Response · Authors · 2024-07-05
> >
> > ------
> > Reviewer Comment:  Not enough information about how baselines were implemented. Specifically for the WML method -- do you rerun meta-training for each individual meta-test task ? My intuition was that WML would perform best (since the weights are learned per meta-test task) but would be too computationally intensive because it's being run for each individual meta-test task. Is your method outperforming WML partially because you aggregate the unlabelled meta-test data and so there is some transfer going on? Can you provide more details about the WML setup ?
> >
> > Response:  For WML, we trained specific model parameters for each meta-test task exactly as described in the original paper[R2]. In Section 5.2, we mention that although both WML and our proposed method use unlabeled datasets from tasks to compute weights, these datasets are different from the meta-test tasks but follow the same task distribution. This setting more closely mirrors real-world scenarios and provides a more rigorous evaluation of the model's adaptability to task distribution shifts. In Section 5.4, we analyze the reasons why our proposed method outperforms WML. One reason is that, in our setting, the tasks used to compute weights do not appear in the meta-test phase, making it difficult for WML to compute accurate weights for each meta-training task. In contrast, our proposed method calculates weights by estimating the density of each meta-training task within both the meta-learning task distribution and the meta-test task distribution, without relying on any specific task. This allows our method to accurately compute weights even when utilizing tasks that conform to the meta-test task distribution but are absent from the meta-test phase.
> >
> > [R2] Weighted meta-learning, arXiv 2020.
> >
> > ------
> >
> > Reviewer Comment:  Why just do 1-step GD on the inner loop ? What happens if you do > 1 step in the meta-learning inner loop ?
> >
> > Response:  For your first question, when training MAML, we followed the experimental setup from [R3], which performed a 1-step gradient descent (GD) in the inner loop. This setup was chosen because the experiments in [R3] demonstrated that MAML could achieve significant performance improvements with just one gradient step. Additionally, this setup helps reduce computational costs. Regarding your second question, the MAML paper [R3] mentioned that although the first gradient step provides the most significant performance improvements, additional gradient steps continue to enhance performance without overfitting. However, the meta-learning process involves gradient updates, and more steps mean higher computational costs. For more details, please refer to the content on page 6 of [R3].
> >
> > [R3] Model-Agnostic Meta-Learning for Fast Adaptation of Deep Networks, PMLR 2021.
> >
> > ------
> >
> > Reviewer Comment:  I'm not sure how to phrase this question but it feels a bit circular, if the Dataset you created for experimentation uses the MMD distance to "simulate task shift" when it's the same distance you use in your method construction. Even if the distinction is more nuanced, I think a more "realistic" task shift could be some combination of dataset from the meta-dataset [1] paper.
> >
> > Response:  To address your concerns, we would like to emphasize that we used three datasets for our experiments. As you mentioned, two of these datasets were used to simulate task shift based on MMD distance. However, the experiments on the tieredImageNet dataset did not simulate task shift using MMD distance. Instead, we followed the method from the original paper [R4] to split the training, validation, and test datasets. In tieredImageNet, this ensures that the training and test classes do not have any semantic overlap, resulting in test classes that are less similar to training classes, making it a more realistic task shift setting. The splitting method for tieredImageNet is discussed in Section 5.1, paragraph 3 of our paper. The experimental results using the tieredImageNet dataset (Table 3) indicate that our method is also effective under more realistic task shift settings.
> >
> >
> > Manuscript Revisions:  To improve clarity and address your concerns, we have updated Section 5.1, paragraph 4 of the revised manuscript to explicitly emphasize that in our experiments with tieredImageNet, we did not split the dataset based on MMD distance but instead followed the method from the original paper[R4].
> >
> > [R4] Meta-learning for semi-supervised few-shot classification, arXiv 2018.
> >
> > ------
> >
> > Reviewer Comment:  The paper does not have a broader impact statement. Please make sure to provide this
> >
> > Response:  Please refer to the Broader Impact Statement section on page 13 of our paper.
> >
> > ------
> >
> > Thank you once again for your valuable feedback, which has helped us improve the clarity and completeness of our manuscript.
> >
> > Best regards

---

> > > ### Comment · Reviewer_ye1d · 2024-07-25
> > > **Response to rebuttal**
> > >
> > > Thanks for the responses. They have clarified my confusions about the paper.
> > > I have updated the evidence and claims section to Yes.

---

### Decision · Action_Editor_MhPT · 2024-08-18

**Recommendation:** Accept with minor revision

**Comment:**

The submission adapts the Weighted Meta-Learning approach of [Cai et al. (2020)](https://arxiv.org/abs/2003.09465) to a setting in which meta-test-time tasks to be labelled are drawn from a shifted distribution relative to meta-training time. The approach requires unlabeled data at meta-training time in order to compute importance weights to re-weight task-wise contributions to the meta-training objective. Results demonstrate an improvement on three few-shot image classification benchmarks: miniImageNet, Omniglot, and tieredImageNet.

The reviewers praised the paper's clear presentation, with ye1d noting the method's simplicity and yGSH highlighting improvements over previous methods. However, they raised concerns about generalizability, with PArP questioning performance in other task shift settings and ye1d requesting clarification on the types of shifts targeted, as well as methodology issues like missing ablations (ye1d). The reviewers suggested addressing these concerns, with PArP emphasizing the need to substantiate the method's effectiveness in various task shift scenarios, ye1d requesting more details on baseline implementations (especially WML), and yGSH asking for clearer experimental setup descriptions. These concerns were sufficiently addressed during the revision process by adjusting claims and including ablation results.

After reviewing the submission in detail, the Action Editor (AE) requests the following minor revisions:

1. The data budget for the proposed IWML method at training time from the comparison methods is not made sufficiently clear. For example, the submission states that, for the given experimental settings, "...30 tasks associated with the unlabeled dataset [were used] to calculate importance." The availability of additional, even unlabeled data, changes the interpretation of the empirical results, i.e., as being the product of algorithm (IWML) *plus* additional data (in the form of unlabelled test tasks). As such, **the authors should make clearer the data made available to each method alongside results included in the empirical evaluation.**

2. The **authors should detail the train-test split enforced for the additional unlabelled data made available at meta-training time,** which is not currently made clear in the paper. Are the tasks/examples used to compute performance (meta-testing accuracy) entirely disjoint from the tasks/examples used to compute importance weights during meta-training, as would be appropriate to prevent fitting to the evaluation set? If not, the authors should rerun the evaluation with this split enforced to determine if the proposed method maintains the reported performance improvement.

3. The **authors should correct the revised claim that "...employing MMD within KDE is also a new technique introduced in this work,"** a claim that was introduced during the revision process. This claim is incorrect. See [Gretton et al. (2012)](https://jmlr.csail.mit.edu/papers/volume13/gretton12a/gretton12a.pdf) for an example of a prior use.

34. Though the AE agrees that three is an audience of individuals interested in algorithms for task shift in few-shot learning who may find interest in this work, this audience is limited by the fact that the evaluation is constructed from simple datasets (variants of the miniImageNet task of [Vinyals et al., 2016](https://arxiv.org/abs/1606.04080)). These datasets are of reduced interest in the transfer learning and few-shot learning literature relative to more complex few-shot benchmarks such as Meta-Dataset ([Triantafillou et al. 2020](https://arxiv.org/abs/1903.03096)) and VTAB ([Zhai et al., 2020](https://arxiv.org/abs/1910.04867)), and task shift benchmarks such as the WILDS dataset, especially v.2.0 with unlabelled data ([Sagawa et al., 2021](https://arxiv.org/abs/2112.05090)). To address this, **the authors should note the lack of demonstrated generalizability to more recent relevant benchmarks in the paper, e.g., in the conclusion or limitations section**.

Subject to these revisions, the AE can recommend acceptance to TMLR.

**Audience:**

Yes

**Claims And Evidence:**

Yes

---

> ### Author Response · Authors · 2024-09-20
>
> Dear Action Editor,
>
> Thank you very much for your insightful comments and valuable feedback. We appreciate the time and effort you have taken to review our manuscript and provide suggestions for improvement.
>
> ------
>
> AE's comment: The data budget for the proposed IWML method at training time from the comparison methods is not made sufficiently clear. For example, the submission states that, for the given experimental settings, "...30 tasks associated with the unlabeled dataset [were used] to calculate importance." The availability of additional, even unlabeled data, changes the interpretation of the empirical results, i.e., as being the product of algorithm (IWML) plus additional data (in the form of unlabelled test tasks). As such, the authors should make clearer the data made available to each method alongside results included in the empirical evaluation.
>
> Response: Thank you for your suggestion regarding the data budget during the meta-training phase. We have updated the second paragraph of Section 5.3 in the paper to clarify the data budget for the proposed IWML and the comparative methods during the meta-training phase. The updated explanation is as follows:
> > During the meta-training phase, since WML and IWML require unlabeled data to compute weights, we used an additional 30 importance estimation tasks consisting solely of unlabeled data for weight calculation. On the other hand, the other comparative methods did not use any additional unlabeled data during the meta-training phase.
>
> Furthermore, in Section 5.4 of the manuscript, we have updated an explanation regarding the performance improvement of IWML. The updated explanation is as follows:
> > The proposed method enables the model to leverage additional unlabeled data from the meta-test task distribution, allowing it to focus more on meta-training tasks that closely align with the meta-test tasks during the meta-training process, resulting in better performance.
>
> This ensures that the performance improvement is attributed to the combination of the algorithm and the use of unlabeled data.
>
> ------

---

> > ### Author Response · Authors · 2024-09-20
> >
> > AE's comment: The authors should detail the train-test split enforced for the additional unlabelled data made available at meta-training time, which is not currently made clear in the paper. Are the tasks/examples used to compute performance (meta-testing accuracy) entirely disjoint from the tasks/examples used to compute importance weights during meta-training, as would be appropriate to prevent fitting to the evaluation set? If not, the authors should rerun the evaluation with this split enforced to determine if the proposed method maintains the reported performance improvement.
> >
> > Response: We appreciate your observation and have clarified that the tasks used for computing importance weights during the meta-training phase are entirely separate from the tasks used to compute meta-test accuracy. To address your concern, we revised the fourth paragraph of Section 5.1 to clarify that these tasks are fully distinct. The updated explanation is as follows:
> > > For both miniImageNet and Omniglot, we simulated a task shift scenario by splitting all classes within each dataset into two clusters based on MMD distance.
> > Specifically, first, we randomly selected two classes. Next, for each class, we calculated its MMD distance to these two selected classes, clustering each class to the one it was closer to. Let denote the two clusters A and B. The meta-training-validation datasets consisted of all classes within cluster A and randomly selected 10\% of the classes within cluster B. The meta-training datasets comprised 70\% of the classes from the meta-training-validation datasets, also selected without replacement. The meta-validation datasets consisted of the remaining 30\% of the classes from the meta-training-validation datasets that were not selected for meta-training. The meta-test datasets was composed of 45\% of the classes randomly selected from cluster B not used for meta-training-validation. The unlabeled datasets for importance estimation was made up of the remaining 45\% of the classes from cluster B not used for meta-training-validation or meta-test. To ensure overlap between the meta-training and meta-test task distributions, the meta-training-validation datasets included all classes from cluster A, as well as 10\% of the classes from cluster B. For tieredImageNet, because the test classes are less similar to the training classes, we followed the original paper's [R1] method to split the training, validation, and test datasets. The unlabeled datasets used for importance estimation consisted of 40 classes randomly selected from the test dataset without replacement. The remaining classes in the test dataset, which were not used for importance estimation, formed the new test dataset. The meta-training datasets comprised all the classes from the training dataset, the meta-validation datasets included all the classes from the validation dataset, and the meta-test datasets contained all the classes from the new test dataset. Since the classes in the meta-test and importance estimation datasets did not overlap, the datasets used for computing importance weights during the meta-training phase were entirely separate from those used for the meta-test.
> >
> > [R1] Meta-learning for semi-supervised few-shot classification, arXiv 2018.
> >
> > ------

---

> > > ### Author Response · Authors · 2024-09-20
> > >
> > > AE's comment: The authors should correct the revised claim that "...employing MMD within KDE is also a new technique introduced in this work," a claim that was introduced during the revision process. This claim is incorrect. See Gretton et al. (2012) for an example of a prior use.
> > >
> > > Response: We appreciate your pointing this out. We have now revised the claim regarding the use of MMD within KDE. Specifically, we removed the statement "...employing MMD within KDE is also a new technique introduced in this work". The updated claim is as follows:
> > > > We propose a task density estimation method by leveraging KDE in conjunction with the MMD distance.
> > >
> > > ------
> > >
> > > AE's comment: Though the AE agrees that three is an audience of individuals interested in algorithms for task shift in few-shot learning who may find interest in this work, this audience is limited by the fact that the evaluation is constructed from simple datasets (variants of the miniImageNet task of Vinyals et al., 2016). These datasets are of reduced interest in the transfer learning and few-shot learning literature relative to more complex few-shot benchmarks such as Meta-Dataset (Triantafillou et al. 2020) and VTAB (Zhai et al., 2020), and task shift benchmarks such as the WILDS dataset, especially v.2.0 with unlabelled data (Sagawa et al., 2021). To address this, the authors should note the lack of demonstrated generalizability to more recent relevant benchmarks in the paper, e.g., in the conclusion or limitations section.
> > >
> > > Response: We acknowledge the AE's suggestion and have included a discussion in the conclusion section (Section 6) regarding the generalizability of the proposed method to more complex benchmarks. The updated explanation is as follows:
> > > > Our evaluation focuses on the miniImageNet, Omniglot, and tieredImageNet datasets, which, although widely used, do not capture the full range of challenges posed by more complex benchmarks such as Meta-Dataset [R2], VTAB [R3], and WILDS [R4]. Future work will aim to address this by applying the method to these benchmarks and assessing its robustness in more challenging scenarios.
> > >
> > > This clarified the limitations of our method in the current experimental setup and outlined potential future work that will explore the application of IWML to more challenging datasets.
> > >
> > > [R2] Meta-dataset: A dataset of datasets for learning to learn from few examples, arXiv 2019.
> > >
> > > [R3] A large-scale study of representation learning with the visual task adaptation benchmark, arXiv 2019.
> > >
> > > [R4] Extending the wilds benchmark for unsupervised adaptation, arXiv 2021.
> > >
> > >
> > > ------
> > >
> > > Thank you once again for your valuable feedback, which has helped us improve the clarity and completeness
> > > of our manuscript.
> > >
> > > Best regards